# Morphological and Transcriptome Analysis of Wheat Seedlings Response to Low Nitrogen Stress

**DOI:** 10.3390/plants8040098

**Published:** 2019-04-15

**Authors:** Jun Wang, Ke Song, Lijuan Sun, Qin Qin, Yafei Sun, Jianjun Pan, Yong Xue

**Affiliations:** 1College of Resources and Environmental Sciences, Nanjing Agricultural University, Nanjing 210095, China; 2016103084@njau.edu.cn; 2Eco-Environmental Protection Research Institute, Shanghai Academy of Agricultural Sciences, Shanghai 201403, China; ke.song@wilkes.edu (K.S.); sunlijuan@saas.sh.cn (L.S.); qinqin@saas.sh.cn (Q.Q.); 2014203039@njau.edu.cn (Y.S.); 3Shanghai Scientific Observation and Experimental Station for Agricultural Environment and Land Conservation, Shanghai Academy of Agricultural Sciences, Shanghai 201403, China; 4Shanghai Environmental Protection Monitoring Station of Agriculture, Shanghai Academy of Agricultural Sciences, Shanghai 201403, China; 5Shanghai Engineering Research Center of Low-Carbon Agriculture (SERLA), Shanghai Academy of Agricultural Sciences, Shanghai 201403, China; 6Shanghai Key Laboratory of Protected Horticultural Technology, Shanghai Academy of Agricultural Sciences, Shanghai 201403, China

**Keywords:** morphological characteristics, transcriptome sequencing, wheat, low nitrogen stress

## Abstract

Nitrogen (N) is one of the essential macronutrients that plays an important role in plant growth and development. Unfortunately, low utilization rate of nitrogen has become one of the main abiotic factors affecting crop growth. Nevertheless, little research has been done on the molecular mechanism of wheat seedlings resisting or adapting to low nitrogen environment. In this paper, the response of wheat seedlings against low nitrogen stress at phenotypic changes and gene expression level were studied. The results showed that plant height, leaf area, shoot and root dry weight, total root length, and number under low nitrogen stress decreased by 26.0, 28.1, 24.3, 38.0, 41.4, and 21.2 percent, respectively compared with plants under normal conditions. 2265 differentially expressed genes (DEGs) were detected in roots and 2083 DEGs were detected in leaves under low nitrogen stress (N-) compared with the control (CK). 1688 genes were up-regulated and 577 genes were down-regulated in roots, whilst 505 genes were up-regulated and 1578 were down-regulated in leaves. Among the most addressed Gene Ontology (GO) categories, oxidation reduction process, oxidoreductase activity, and cell component were mostly represented. In addition, genes involved in the signal transduction, carbon and nitrogen metabolism, antioxidant activity, and environmental adaptation were highlighted. Our study provides new information for further understanding the response of wheat to low nitrogen stress.

## 1. Introduction

Wheat is one of the most widely grown cereal crops all over the world [1]. However, low utilization rate of nitrogen (N) fertilizer severely limits the yield and quality of wheat [2]. Excessive application of nitrogen fertilizer is one of the main ways to ensure crop yield and quality, yet plants can only use about 30% to 40% of the applied nitrogen fertilizer. No less than 40% of the nitrogen fertilizer applied is lost by leaching into the groundwater, lakes, rivers and atmosphere, giving rise to severe pollution [3]. In order to solve this problem, the technology of ‘reducing fertilizer and increasing efficiency’ has been popularized in our country. Recent studies have shown that insufficient nutrient supply has a serious impact on plant growth [4,5,6]. Evidence from Jeuffroy et al. indicated that nitrogen deficiency of winter wheat generally can result in slow growth, fewer tillers, and yellowish leaves [7]. The negative effects of low nitrogen on the formation of wheat root morphology have been discussed, including the decreased root length, root number, root surface area, and root dry weight [2,8]. Furthermore, Rose et al. suggested that the developed root architecture has stronger nitrogen uptake capacity, such as greater root length and root surface area [9]. Understanding the morphological response characteristics of wheat seedlings under low nitrogen stress is of critical importance for agricultural fertilization and selection of resistant varieties.

In recent years, transcriptome profiling using next-generation sequencing technologies has been used to study the transcription of genes and the regulation of transcriptional at the overall level [10]. Transcriptome analysis based on Illumina’s RNA-sequencing platform in order to explore gene expression in response to nitrogen nutrition stress in plants has been carried out. Wan et al. revealed that wheat amino acid transporters play a vital role in nitrogen transport, response to abiotic stress, and development based on transcriptome analysis [11]. Dai et al. studied how the accumulation of wheat grain storage protein is regulated during grain development in response to nitrogen supply by using transcriptome profiling [12]. Asparagine is considered to be an ideal nitrogen transport molecule, as it plays a major in nitrogen uptake by plant roots [13]. Previous works have found asparagine synthetase genes (AsnS) in *Arabidopsis* [14] and maize [15]. Curci et al. showed that AsnS genes were down-regulated in durum wheat roots and leaves under nitrogen stress [1]. In addition, when plants grow nitrogen-free condition, the genes involved in nitrogen compound metabolism, carbon metabolism, amino acid metabolism, and photosynthesis were down-regulated in roots and leaves [16,17].

Although great progress has been done on the adaptation mechanisms of plants to abiotic stress, little research is available on wheat seedlings response to low nitrogen stress. In this work, besides studying morphological changes between low nitrogen stress and control samples, we focused on transcriptome analysis of wheat seedlings under low nitrogen stress using high-throughput sequencing. Our results provide a basis for understanding wheat’s response to low nitrogen stress and the molecular mechanisms that underlie it.

## 2. Results

### 2.1. Low Nitrogen Stress Affects Wheat Seedlings Morphology

Wheat seedlings exposure to N- produced a significant growth inhibition compared to control (CK), as observed on plant height, leaf area, shoot dry weight, root dry weight, total root length, and total root number. N- led to a significant (*p* < 0.05) decrease in plant height, leaf area, shoot dry weight, root dry weight, total root length, and total root number by 26.0, 28.1, 24.3, 38.0, 41.4 and 21.2 percent, respectively, compared to CK (Table 1). These results suggest that wheat seedlings are highly sensitive to low nitrogen environments, and low nitrogen stress seriously affected the growth of wheat.

### 2.2. Overview of Transcriptome Sequencing Results

The genome-wide transcriptional response to low nitrogen stress in wheat seedlings was investigated by high-throughput RNA-seq. Seedling samples, CK and N-, were sequenced using an Illumina HiSeq platform. Approximately 40.83 to 45.10 million 150 bp paired-end clean reads were obtained from leaf CK and N- samples, respectively, while root CK and N- samples engendered 40.73 to 43.72 million reads, respectively, after adapter trimming and filtering low-quality reads (Table 2). The average leaf Q20, Q30, and GC (Base G + Base C) contents were 95.39%, 89.06%, and 55.00%, respectively. Similarly, the average root Q20, Q30, and GC contents were 95.12%, 88.67%, and 54.00% respectively, with the clean reads of Q20 occupying over 95% of the total reads. These findings attest to the fine quality of the sequencing results. The respective mapped reads information between CK and N- leaf samples was: 84.43% and 84.03% total mapped; 8.06% and 6.50% multiple mapped; 91.94% and 93.50% uniquely mapped, respectively. Between CK and N- root samples 71.01% and 77.41% were total mapped, 6.76% and 5.64% were multiple mapped, while 93.24% and 94.36% were uniquely mapped (Table 2). The transcriptome data was deemed suitable for subsequent analysis.

### 2.3. Low Nitrogen Stress Affects Genes Expression in Wheat Seedlings

Compared with the control, a total of 2265 DEGs were found in roots under low nitrogen stress, of which 1688 genes were up-regulated and 577 genes were down-regulated (Figure 1, Appendix A). In a total of 2083 DEGs in leaf transcripts, 505 genes were up-regulated and 1578 genes were down-regulated (Figure 1, Appendix A). We observed that a larger number of genes were up-regulated in roots, and a larger number of genes were down-regulated in leaves. Overall, the total number of DEGs in roots were higher than in leaves.

### 2.4. GO Enrichment Analysis of DEGs

Gene ontology (GO) enrichment analysis was carried out to further characterize the main biological functions of DEGs in wheat seedlings under low nitrogen stress. All the DEGs can be divided into three categories, including biological process, molecular function, and cellular component. Furthermore, the three categories could further be divided into 45 subcategories in the leaf, of which 28 subcategories were significantly (*p* < 0.05) enriched (Figure 2; Appendix A). There were ten, ten, and six enriched subcategories belonging to the categories of biological process, molecular function and cellular component, respectively. In biological process, the ‘oxidation reduction process’ was the most enriched subcategory. ‘Oxidoreductase activity’ was the most enriched subcategory in the molecular function. Among the cellular components, the three most enriched subcategories were ‘cell’, ‘cell part’, and ‘organelle’. Other significantly enriched subcategories are shown in Appendix A.

Moreover, these three categories could be further divided into 38 subcategories in the root, of which 20 subcategories were significantly (*p* < 0.05) enriched (Figure 3; Appendix A). There were ten and nine enriched subcategories belonging to the categories of biological process and molecular function, respectively. Only the ‘extracellular region’ was significantly enriched in the cellular component. ‘Organic acid catabolic process’ and ‘carbon-nitrogen lyase activity’ were the most highly enriched in each of the biological process and molecular function categories, respectively. Other significantly enriched subcategories are shown in Appendix A.

### 2.5. Kyoto Encyclopedia of Genes and Genomes (KEGG) Enrichment Analysis of DEGs

In order to assign DEGs to cellular pathways, pathway enrichment analysis based on KEGG was performed. The DEGs significantly enriched pathways were calculated using hypergeometric distribution based on the whole genome. There are five KEGG pathway categories: Cellular processes, environmental information processing, genetic information processing, metabolism, and organismal systems. ‘Signal transduction’ was the only item enriched in environmental information processing in the leaf. In regard to metabolism, ‘carbohydrate metabolism’ was the most highly overrepresented, followed by ‘metabolism of terpenoids and polyketides’, ‘amino acid metabolism’, ‘lipid metabolism’, ‘energy metabolism’, ‘metabolism of cofactors and vitamins’, ‘biosynthesis of other secondary metabolites’, and ‘metabolism of other amino acids’. ‘Environmental adaptation’ and ‘nervous system’ were the only items enriched in organismal systems. No enrichment was found in cellular processes and genetic information processing (Figure 4).

Similar KEGG pathway enrichments were found in roots. ‘signal transduction’ was the only enriched process in environmental information processing. In the category of metabolism, most of the pathways were highly overrepresented, including ‘biosynthesis of other secondary metabolites’, ‘amino acid metabolism’, ‘lipid metabolism’, ‘energy metabolism’, ‘xenobiotics biodegradation and metabolism’, ‘metabolism of other amino acids’, ‘carbohydrate metabolism’ and ‘metabolism of terpenoids and polyketides’. In the category of organismal systems, ‘environmental adaptation’ was the mostly overrepresented, followed by ‘nervous system’, ‘endocrine system’, ‘sensory system’, and ‘digestive system’. No significant enrichment was found in cellular processes and genetic information processing (Figure 5).

### 2.6. Validation of RNA Sequencing Data by Quantitative Real-Time PCR

A total of eight DEGs (four in leaf and four in root) were randomly selected and validated by quantitative real-time PCR (qRT-PCR) to confirm the reliability of our sequencing data. These included genes encoding photosystem II PPD protein 3, MYB transcription factor 78, glycine cleavage system p protein, and catalase in leaves, and WRKY transcription factor, asparagine synthetase, peroxidase, and MYB 33 in roots. The results showed that the average expression levels of four genes in leaf were significantly down-regulated under low nitrogen stress, whereas the average expression levels of four genes in root were significantly up-regulated under low nitrogen stress (Figure 6). The results showed that the expression of these genes measured by qRT-PCR was in good agreement with the RNA-seq results. Consequently, the RNA-seq data we obtained were trustworthy.

## 3. Discussion

### 3.1. Response to Low Nitrogen Stress by Morphological Changes

Nitrogen is an essential nutrient for plant growth and development [17]. Insufficient nitrogen supply adversely influences morphology, limits growth, and decreases biomass in wheat [18,19]. Most plants manifest prominent changes in their leaves and roots when grown under low phosphorus or nitrogen conditions. Plants rely on morphological changes to adapt to nutrient stress, a common finding on plants grown under nutrient stress conditions [20,21,22]. Our results further confirmed that low nitrogen stress inhibited wheat growth. We recorded a remarkable response to low nitrogen stress in wheat seedlings. Low nitrogen treatment significantly decreased wheat seedlings growth parameters, including plant height, leaf area, shoot and root dry weight, and total root length and number. The changes in these parameters suggested that low nitrogen stress has a serious impact on wheat seedlings growth.

### 3.2. Potential DEGs Play Important Roles in Low Nitrogen Tolerance in Wheat Seedlings

Abiotic stress triggers dramatic molecular responses in plants. In recent years, the molecular mechanism of plant response to abiotic stress has attracted wide attention [23,24,25,26]. In order to obtain the molecular mechanism of wheat seedlings response to low nitrogen stress, RNA-seq analysis was performed. 4348 differentially expressed genes were obtained from the whole wheat seedling in all, and among which 2265 genes (1688 up-regulated and 577 down-regulated) were from roots, and 2083 genes (505 up-regulated and 1578 down-regulated) were from leaves. Our results showed that the gene expression of wheat seedlings changed greatly under low nitrogen stress. Changes in gene expression can lead to changes in the corresponding biological processes. GO enrichment analysis is helpful to highlight the main biological processes in response to stress environment. For example, ‘nitrogen compound metabolism’, ‘carbon metabolism’, and ‘photosynthesis’ were mostly enriched in durum wheat under nitrogen starvation [17]. ‘Metabolic process’, ‘cellular process’, and ‘transport’ were enriched in rice roots and shoots under nitrogen-free conditions [16]. Our results detected ‘oxidation-reduction process’ and ‘metabolic process’ as highly enriched in wheat leaves subjected to low nitrogen stress. Genes involved in ‘carbon–nitrogen lyase activity’ and ‘organic acid catabolic process’ were significantly enriched in wheat seedlings roots. Bi et al. indicated that DEGs associated with these processes might play essential roles in *Arabidopsis thaliana* adaptation to low nitrogen stress [27]. Therefore, the response of these biological process genes enhances the adaptability of wheat to low nitrogen stress.

### 3.3. Amino Acid Metabolism, Lipid Metabolism, Energy Metabolism, and Signal Transduction Pathway Play Important Roles Under Low Nitrogen Stress

KEGG pathway analysis can help us to further understand the biological functions of genes and how these genes interact [28]. Previous studies have revealed that many genes participate in nitrogen deficiency or nitrogen-free condition resistance via various amino acid metabolism, lipid metabolism, energy metabolism, and signal transduction pathways [29,30,31]. For example, DEGs associated with amino acid metabolism pathway were mostly represented in sorghum roots under low nitrogen stress, which play a key role in nitrogen uptake and transformation [32]. Protein kinases (PK) are widely involved in the signal transduction and were shown to respond to nitrogen deficiency in *Arabidopsis thaliana* roots and leaves [17]. In this study, genes related to ‘amino acid metabolism pathway’, ‘lipid metabolism pathway’, ‘energy metabolism pathway’, and ‘signal transduction pathway’ were also identified both in leaves and roots. These results support previous findings that these pathways can form a close-knit signaling network and play vital roles in low nitrogen stress tolerance in plants [28].

### 3.4. Some Candidate Genes for Plant Low Nitrogen Stress Tolerance Breeding

PsbP protein is an extrinsic component of photosystem II (PSII) and participates in crucial processes such as calcium-ion binding and photosynthesis [33,34]. Studies have shown that genes encoding PsbP protein can serve as signal response factors to various unfavorable external environments [35,36]. In our study, PsbP domain-containing protein 3 (PPD3) gene, a member of the PsbP gene family, was predominantly down-regulated in wheat leaves under low nitrogen stress., Before this, no studies have found that PPD3 gene is regulated in wheat under low nitrogen or nitrogen-free stress. Our results provide a reference for future research.

Transcription factors (TFs) are important factors involved in abiotic stress regulation in plants, and, among them, MYB, WRKY, bHLH, and bZIP families were identified responding to nitrogen stress [37,38]. For example, genes encoding WRKY TFs were up-regulated in rice under nitrogen deficiency [16], MYB TFs were up-regulated in durum wheat under nitrogen starvation [17], and also revealed that NF-Y TFs were induced in wheat under low nitrogen conditions, which play roles in nitrogen uptake and grain yield [39]. Some TFs regulate the pivotal cell process in the response to nitrogen deficiency. Zhang et al. reported that several MYB TFs are associated with the cell development and cell cycle in wheat during nitrogen stress [40]. In this paper, MYB and WRKY were the most prominent. In regard to WRKY family genes, 17 genes were up-regulated in roots under low nitrogen stress, 40 MYB family genes were detected in both the leaves and roots. They were down-regulated in leaves, but up-regulated in roots..Our results suggest that these transcription factors detected in response to low nitrogen stress may contribute to the adaptation or resistance of wheat seedlings to low nitrogen condition.

Asparagine (Asn) play a vital role in nitrogen metabolism, transport, and storage [41]. Asparagine biosynthesis is catalyzed by asparagine synthase (AsnS) in higher plants [42]. Recent studies showed that the AsnS gene family also serves as a response factor to nitrogen and other abiotic stresses [17]. Asparagine synthetase 1 (AS1), a member of the AsnS family, was shown in rice to be expressed mainly in roots in an NH_4_^+^-dependent manner [43]. In the present work, several genes belonging to the AsnS gene family were mostly up-regulated in wheat roots under low nitrogen stress. The significant differential expression of AsnS genes in wheat roots under low nitrogen condition may contribute to the synthesis of asparagine, and promote the assimilation and distribution of nitrogen.

Reactive oxygen species (ROS) often accumulates in plants under various abiotic stresses, which leads to the oxidative damage of many cell structures and components [44,45]. In order to protect themselves from oxidative damage, plants have developed an antioxidant defense system consisting of a variety of enzymes. Superoxide dismutase (SOD), peroxidases (POD), and catalases (CAT) are common antioxidant enzymes. Lian et al. have reported that antioxidant defense systems can be induced under low nitrogen stress, and improve the adaptability to low nitrogen environment, by increasing the gene expression levels of SOD, POD, and CAT in rice [46]. Bi et al. also found that several detoxification-associated genes were detected in *Arabidopsis* under nitrogen stress [27]. In the present study, plenty of POD genes were up-regulated in wheat roots under low nitrogen stress, and several CAT genes were down-regulated in leaves. These genes might improve the adaptability of wheat seedling to low nitrogen environments. 

## 4. Material and Methods

### 4.1. Plant Material and Growth Conditions

For this study, wheat cultivar Wanmai No. 52 grown in large areas in China, was selected. The wheat seedlings were hydroponically cultured in a greenhouse. After selecting full and uniform seeds, they were sterilized with ethanol (75% *v*/*v*) for one min and then washed three times in distilled water, soaked in distilled water and placed in an artificial climate incubator at 25 °C for 24 h without light. Uniformly germinated seeds were selected and placed on a moistened germination paper, and then cultured in a greenhouse at 25 ± 3 °C. At about 5 cm growth height, the robust seedlings were transferred to plastic boxes containing a proper volume of hydroponic nutrient solution (pH 6.0), and were grown afterwards in an incubator at 20/15 °C (day/night) under a 12 h photoperiod until they showed two fully expanded leaves. During the growth period, seedlings were sprayed with distilled water on time. The growth of wheat is shown in Appendix A.

### 4.2. Experimental Design

The basic nutrient solution was an improved Hoagland nutrient solution. The major elements of nutritional solution are as follows: 4 mM Ca(NO_3_)_2_·4H_2_O, 5 mM KNO_3_, 1 mM NH_4_NO_3_, 1 mM KH_2_PO_4_, 2 mM MgSO_4_ × 7H_2_O, 0.02 mM FeSO_4_ × 7H_2_O + EDTA(Na). The micronutrients of nutritional solution are as follows: 0.005 mM KI, 0.1 mM H_3_BO_3_, 0.15 mM MnSO_4_, 0.05 mM ZnSO_4_, 0.001 mM Na_2_MoO_4_, 0.15 × 10^−3^ mM CuSO_4_, 0.19 × 10^−3^ mM CoCl_2_. Two-leaf stage seedlings were transferred to different conditions: Standard total nitrogen nutrition (Ca(NO_3_)_2_·4H_2_O+KNO_3_ + NH_4_NO_3_ 2 mM, labeled CK), and low-nitrogen stress (Ca(NO_3_)_2_·4H_2_O + KNO_3_ + NH_4_NO_3_ 0.2 mM, labeled N-). Except for the total nitrogen concentration, the other components of the solution under low-nitrogen stress were identical to those of the control. Each treatment was applied to 18 plants and repeated 3 times. The incubator was placed in a greenhouse with a 12 h photoperiod and the nutrient solution was renewed every three days. After ten days of treatments, roots and leaves were sampled from CK and N- wheat seedlings, and immediately frozen in liquid nitrogen, and then refrigerate it in the refrigerator with −80 °C until RNA extraction.

### 4.3. Determination of Morphological Parameters

Morphological parameters (plant height, leaf area, shoot dry weight, root dry weight, total root length and number) were determined from harvested wheat seedlings. Plant height and leaf area were determined following Wan [47]. Leaf area was calculated according to the following formula:
Leaf area (cm^2^) = leaf length × leaf width/1.2

Roots were scanned with a scanner, and images were analyzed using Win Rhizo to determine total root length and total root number according to Boris [48]. Seedlings dry weight was determined according to Qiu [49].

### 4.4. Isolation of Total RNA, cDNA Library Construction, and Illumina Sequencing

Total RNA was extracted from roots and leaves by the TRIzol Reagent (Invitrogen, Carlsbad, CA, USA) according to the manufacturer’s instructions. Agilent 2100 Bioanalyzer (Agilent Technologies, Santa Clara, CA, USA) was used to determine RNA concentration and purity. RNA integrity was examined by running gel electrophoresis in 1% (w/v) agarose gel (Invitrogen, CA, USA). The next step is to enrich mRNA from total RNA using Oligo (dT)-conjugated magnetic beads. mRNA was enriched from total RNA using mRNA fragmentation was conducted by ion interruption, giving rise to mRNA fragments of about 200 bp. The first strand of cDNA was synthesized using reverse transcriptase and random primers using the above fragments as templates, and the second strand was synthesized using DNA polymerase I and RNase H. The end of the fragment was connected with the adapter. Then the products were purified and concentrated by polymerase chain reaction (PCR) to form the final cDNA libraries. The libraries were paired-end (PE)-sequenced using next-generation sequencing based on the Illumina HiSeq platform. Sequencing was conducted by Shanghai Personal Biotechnology Co., Ltd. (Shanghai, China). The raw reads were submitted to the NCBI Sequence Read Archive (accession number: PRJNA528563).

### 4.5. Data Filtering and Mapping of Illumina Reads

To obtain high-quality reads for subsequent analysis, FastQC quality control tool (http://www. bioinformatics.babraham.ac.uk/projects/fastqc/) was used to evaluate the quality of the quality of RNA-seq raw reads. The reads were then filtered using Cutadapt. The process included removing adapter sequences and trimming the bases with a quality score less than Q20 using a 5-bp 3′ to 5′ window. Final reads that are less than 50 bp in length or do not require bases are discarded. A reference genome index was established using Bowtie2 [50], and clean reads were mapped to the reference genome (http://www.ensembl.org/index.html) using Tophat2 [51].

### 4.6. Enrichment Analysis of DEGs

The DESeq R package was performed to analyze the differential expression in each tissue. At first, we mapped high-quality reads data to the reference genome of wheat (http://www.ensembl.org/) to calculate the number of reads mapped to each gene in each sample. Then these raw read counts were normalized to reads per kilo bases per million reads (RPKM). We use a false discovery rate of 0.05 as the threshold for judging the significance of DEGs. We performed enrichment analysis using GO (http://geneontology.org/) and the KEGG (http://www.genome.jp/kegg). DEGs terms and pathways were calculated using a hypergeometric distribution algorithm with wheat reference genome (Appendix A) (http://www.ensembl.org/) as background.

### 4.7. Quantitative Real-Time PCR Validation

In order to validate the reliability of DEGs obtained from RNA-seq, four DEGs were randomly selected from leaves and roots, respectively, for qRT-PCR analysis. *RLI* (Ta2776) was used as an internal control. Gene-specific primer pairs were designed using Primer Premier 5.0 software (Premier Biosoft International), and primer information is shown in Appendix A. Total RNA from each tissue was extracted as described above. Two micrograms RNA was reverse-transcribed into cDNA using the iScriptTM advanced cDNA Synthesis Kit (Promega, Madison, WI, USA) following RNase-free DNase I (Promega, Madison, WI, USA) treatment. Standard curve for each gene was prepared with several dilutions of cDNA. The qRT-PCR was carried out using SYBR^®^ Green PCR Master Mix (Roche, CH) in a Rotor-Gene 3000 Real Time system (Qiagen, Hilden, Germany). Quantitative PCR reactions cycling conditions were performed as follows: 95 °C for 5 min, followed by 40 cycles at 95 °C for 15 s, 60 °C for 30 s. The relative expression value of the different genes was calculated using 2^−^^△△Ct^ method [52]. The experiment was performed using three biological replicates.

### 4.8. Statistical Analysis

Statistical analysis of plant morphological data was conducted using SPSS 19.0 (IBM, Chicago, IL, USA). Statistical results were obtained by one-way analysis of variance (ANOVA) followed by Tuckey’s test to evaluate significant treatment effects at significance of *p* < 0.05. Data presented are means ± standard errors.

## 5. Conclusions

Our study provides a more comprehensive understanding of the morphological changes and the DEGs in wheat roots and leaves under low nitrogen stress. It was observed that plant height, leaf area, shoot dry weight, root dry weight, total root length and total root number of wheat seedlings were decreased significantly in under nitrogen stress. 2265 and 2083 DEGs were detected in roots and leaves, respectively.In regard to up-regulation and down-regulation, 1688 genes were up-regulated, and 577 genes were down-regulated in roots, while 505 genes were up-regulated, and 1578 genes were down-regulated in leaves. Furthermore, the classification functional enrichment and metabolic pathways of DEGs were shown in this paper. Several key genes and TFs involved in the signal transduction, carbon and nitrogen metabolism, antioxidant activity, and environmental adaptation were identified. Our results will provide useful information to the further research about wheat response to low nitrogen stress.

## Figures and Tables

**Figure 1 plants-08-00098-f001:**
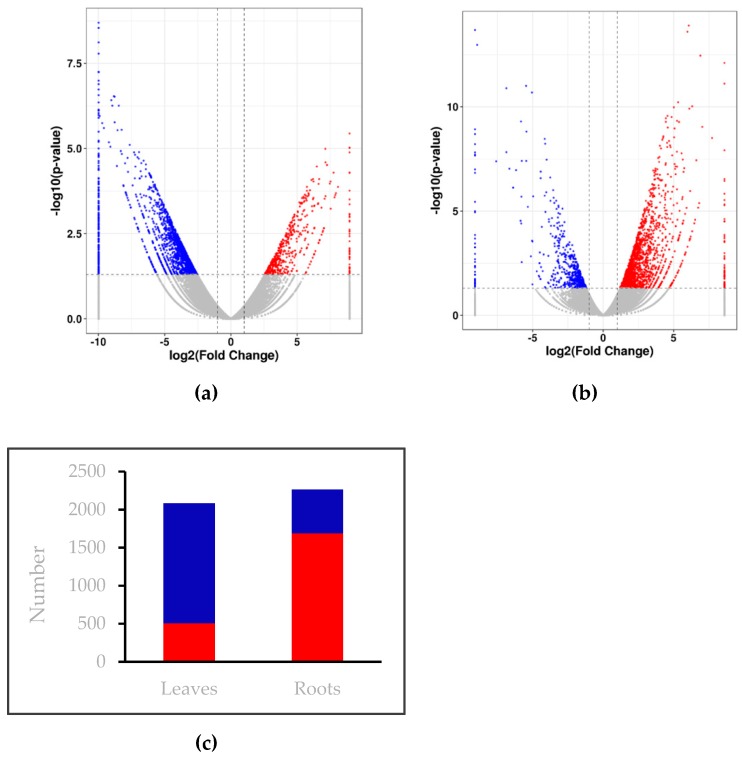
Volcano Plot of differentially expressed genes (DEGs) between the control and low nitrogen stress of wheat seedlings leaves (**a**) and roots (**b**). The two vertical dotted lines are twice of the difference threshold, and the horizontal dotted line represents *p*-value is 0.05. The red dots indicate the up-regulated genes in this group, the blue dots indicate the down-regulated genes in this group, and the gray dots indicate the non-significant differentially expressed genes. (**c**) Number of genes up-and down-regulated in leaves and roots. The red column indicate the number of up-regulated genes, while the blue column indicate the number of down-regulated genes.

**Figure 2 plants-08-00098-f002:**
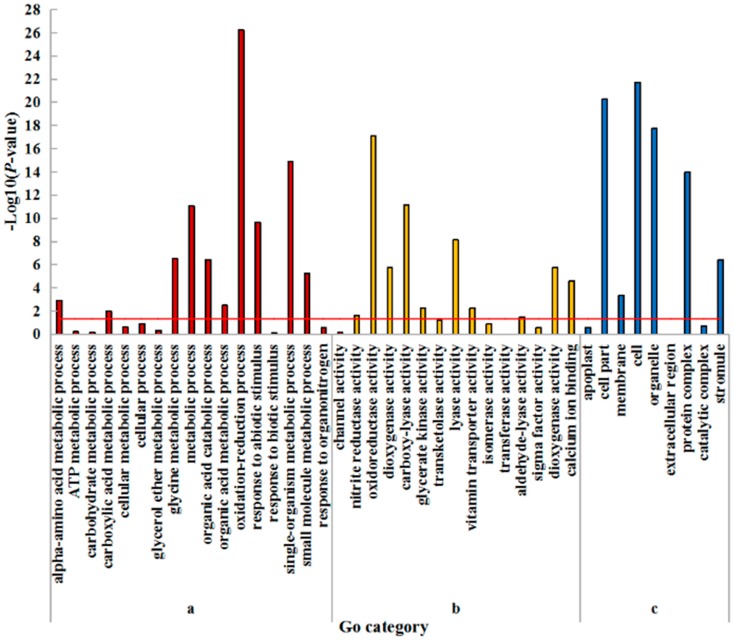
GO enrichment analysis of DEGs in leaves of wheat seedlings. (**a**) Biological process; (**b**) Molecular function; (**c**) Cellular component. The hypergeometric distribution calculates the *p*-value which determines the significance of enrichment, and the red line represents the *p*-value of 0.05.

**Figure 3 plants-08-00098-f003:**
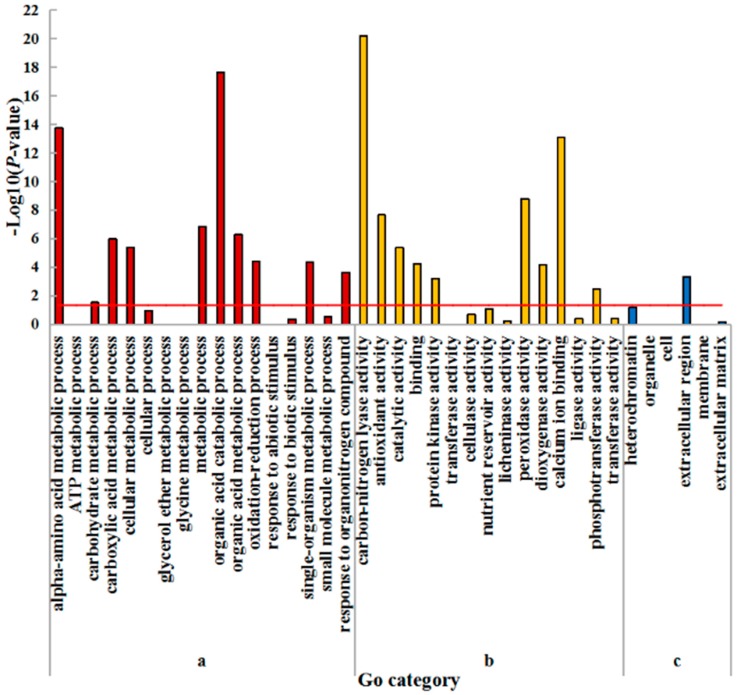
GO enrichment analysis of DEGs in roots of wheat seedlings. (**a**) Biological process; (**b**) Molecular function; (**c**) Cellular component. The hypergeometric distribution calculates the *p*-value which determines the significance of enrichment, and the red line represents the *p*-value of 0.05.

**Figure 4 plants-08-00098-f004:**
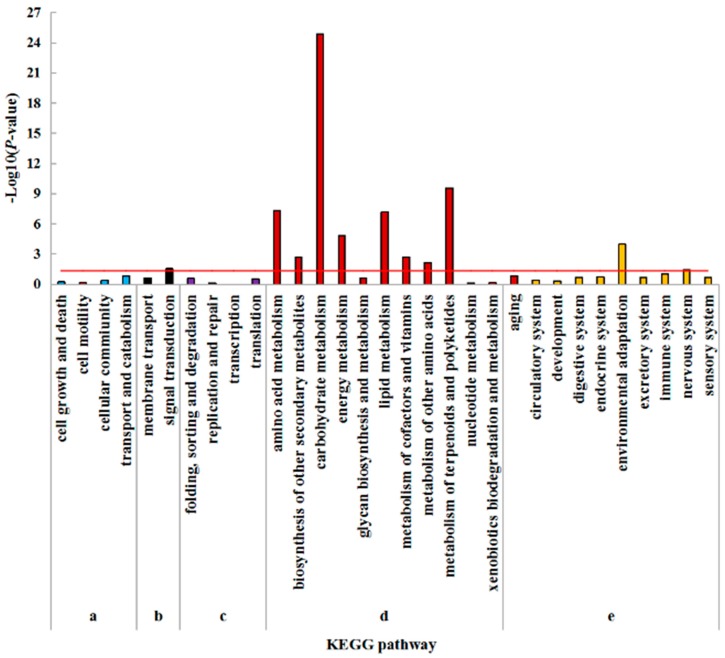
KEGG pathway enrichment analysis of DEGs in leaves under N- compared with CK. (**a**) Cellular processes; (**b**) Environmental information processing; (**c**) Genetic information processing; (**d**) Metabolism; (**e**) Organismal systems. The hypergeometric distribution calculates the *p*-value which determines the significance of enrichment, and the red line represents the *p*-value of 0.05.

**Figure 5 plants-08-00098-f005:**
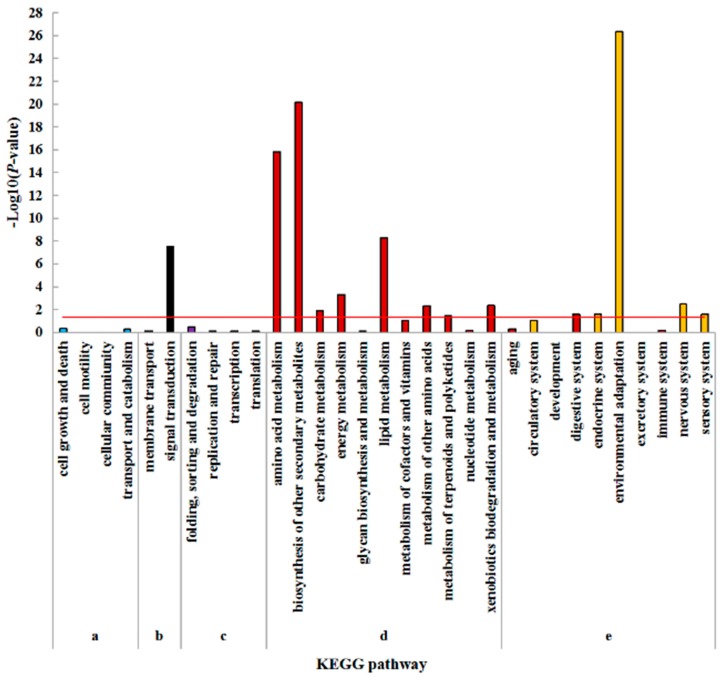
KEGG pathway enrichment analysis of DEGs in roots under N- compared with CK. (**a**) Cellular processes; (**b**) Environmental information processing; (**c**) Genetic information processing; (**d**) Metabolism; (**e**) Organismal systems. The hypergeometric distribution calculates the *p*-value which determines the significance of enrichment, and the red line represents the *p*-value of 0.05.

**Figure 6 plants-08-00098-f006:**
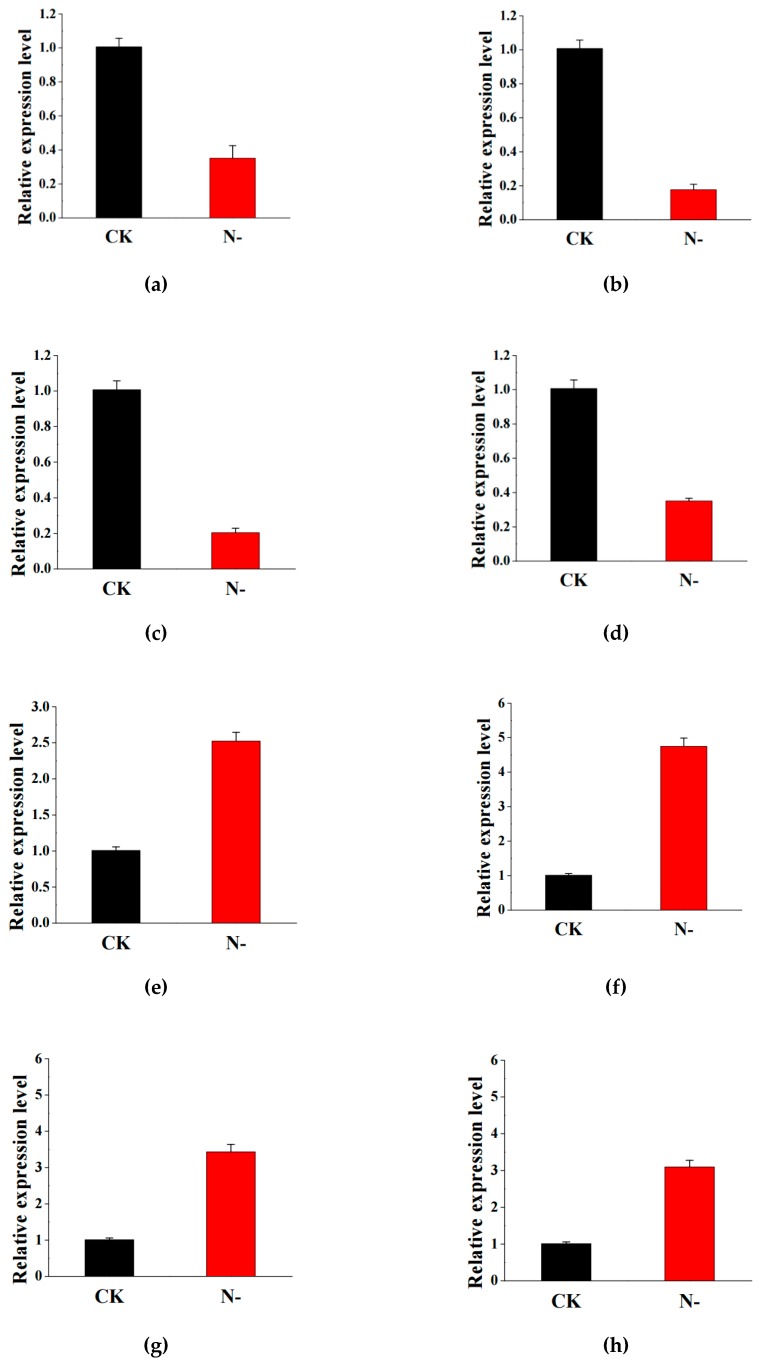
The relative gene expression of eight randomly selected genes examined by qRT-PCR analysis. Different treatments represent: The control (CK), low nitrogen stress (N-). Genes are (**a**) PsbP domain-containing protein 3; (**b**) MYB transcription factor 78; (**c**) Glycine cleavage system p protein; (**d**) Catalase; (**e**) WRKY transcription factor; (**f**) Asparagine synthetase; (**g**) Peroxidase; (**h**) MYB 33. Data represent the mean ± SE (*n* = 3).

**Table 1 plants-08-00098-t001:** Effects of low nitrogen stress on plant height, leaf area, shoot dry weight, root dry weight, total root length, and total root number of wheat seedlings. Different treatments represent: The control (CK), low nitrogen stress (N-). The data are from the average of 15 seedlings, for each parameter, mean values (±standard error) are presented.

Treatment	Plant Height (cm)	Leaf Area (cm^2^/Plant)	Shoot Dry Weigh (mg/Plant)	Root Dry Weigh (mg/Plant)	Total Root Length (cm/Plant)	Total Root Number
CK	26.58 ± 0.60 ^a^	14.70 ± 0.17 ^a^	30.77 ± 0.27 ^a^	23.17 ± 0.57 ^a^	57.83 ± 1.31 ^a^	8.50 ± 0.26 ^a^
N-	19.67 ± 0.56 ^b^	10.57 ± 0.24 ^b^	23.29 ± 0.32 ^b^	14.37 ± 0.76 ^b^	33.90 ± 1.25 ^b^	6.70 ± 0.03 ^b^

Note: Different letters (a,b) indicate that there are significant differences at 0.05 level according to Tuckey’s test.

**Table 2 plants-08-00098-t002:** Summary of RNA-seq data and reads mapping. Different treatment represents: The CK and N- of leaf, the CK and N- of root.

Sample		Raw Reads	Clean Reads	Q20 (%)	Q30 (%)	GC (%)	Total Mapped	Multiple Mapped	Uniquely Mapped
Leaf	CK	45,633,102	45,096,752	95.24	88.76	55.38	38,073,404 (84.43%)	3,068,994 (8.06%)	35,004,410 (91.94%)
	N-	41,236,032	40,827,914	95.53	89.35	54.62	34,306,913 (84.03%)	2,229,665 (6.50%)	32,077,248 (93.50%)
Root	CK	41,250,032	40,726,872	95.24	88.94	53.08	28,921,761 (71.01%)	1,953,811 (6.76%)	26,967,950 (93.24%)
	N-	44,308,466	43,721,986	95.00	88.39	54.92	33,846,419 (77.41%)	1,908,786 (5.64%)	31,937,633 (94.36%)

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
