# Peer review of "Morphological and Transcriptome Analysis of Wheat Seedlings Response to Low Nitrogen Stress"

_plants, 2019, doi:10.3390/plants8040098_

Reviewer 1 Report

Corrections needed to be made in the manuscript:

1. The Introduction and Discussion sections should be supplemented with references to the works on wheat transcriptome under different nitrogen regimes, e.g. Dai et al., Plant J., 2015, 83:326-343; Zheng et al., Sci. Rep., 8:11928; Wen et al., Int. J. Mol. Sci., 2018, 19:3417.

2. Table S4 contains the composition of the basic nutrient solution.  The differences in the low nitrogen solution should be explained.

3. In the Supplementary material the table numbers are incorrect: in text of the manuscript Table S1 is mentioned after Tables S2-S4.

Author Response

Response to Reviewer 1 Comments

Point 1: The Introduction and Discussion sections should be supplemented with references to the works on wheat transcriptome under different nitrogen regimes, e.g. Dai et al., Plant J., 2015, 83:326-343; Zheng et al., Sci. Rep., 8:11928; Wen et al., Int. J. Mol. Sci., 2018, 19:3417.

Response 1: Thank you very much for your comments. I have made some improvements in this aspect of this article. Please refer to the introduction and discussion sections of the manuscript.

Point 2: Table S4 contains the composition of the basic nutrient solution. The differences in the low nitrogen solution should be explained. 

Response 2: In combination with reviewer 3's point 4, these have been revised in the article. Please refer to section 5.2 on page 13 of the manuscript.

Thank you very much.

Point 3: In the Supplementary material the table numbers are incorrect: in text of the manuscript Table S1 is mentioned after Tables S2-S4.

Response 3: The table numbers of supplementary material has been corrected. Please check the table. Thank you very much.

Reviewer 2 Report

The authors studied the nitrogen deficit effects on morphological and transcriptional levels. However, the statistical analyses in the manuscript are not convincing. 

     Table 1: Duncan’s multiple range test lacks robustness. To get a convincing result, I suggest authors use Tuckey’s test. It does not make sense to test the phenotype differences between (CK)-(N) and CK and N, respectively. The authors only need to test the difference of the phenotypes between CK and N group. Please use the small letters instead of the capital letters to label the significant difference.

2.      M&M 5.6: Please use a false discovery rate of 0.05 as the significant threshold when screening DEGs significance because there are multiple tests so that they need a FDR corrected p-value.

3.     M&M 5.6: Which wheat genome assembly information did the authors use as enrichment analysis background?

Author Response

Response to Reviewer 2 Comments

Point 1: Table 1: Duncan’s multiple range test lacks robustness. To get a convincing result, I suggest authors use Tuckey’s test. It does not make sense to test the phenotype differences between (CK)-(N) and CK and N, respectively. The authors only need to test the difference of the phenotypes between CK and N group. Please use the small letters instead of the capital letters to label the significant difference.

Response 1: Thank you very much for your valuable comments. Table 1 has been revised in accordance with your proposal. Please check it.

Point 2: M&M 5.6: Please use a false discovery rate of 0.05 as the significant threshold when screening DEGs significance because there are multiple tests so that they need a FDR corrected p-value.

Response 2: It has been modified. Please refer to lines 341-342 on page 14 of the manuscript. Thank you very much.

Point 3: M&M 5.6: Which wheat genome assembly information did the authors use as enrichment analysis background?

Response 3: The genome assembly information of wheat has been added to the article. Please refer to lines 345-346 on page 14 of the manuscript and Table S3. Thank you very much.

Reviewer 3 Report

The work describes the physiological and transcriptional responses of wheat seedlings to growth in low N condition. Some major revisions are necessary before the publication in Plants.

-       Introduction: the description of use of RNAseq in order to characterize the plants responses to stresses is unnecessary for the aim of the work. The authors have to improve the introduction with information concerning physiological and molecular aspects of N nutrition in plant with particular attention to low N condition. Some transcriptional works focused on transcriptional responses to low N condition was performed in plant species as arabidopsis and maize.

-       The authors repeated in the discussion many results already described. In addition, it is more important to discuss the results on the basis of the molecular events involved into N nutrition rather than the number of differentially expressed transcripts as compared to other stresses. Discussion has to be rewritten and improved.

-       Differentially expressed transcripts were not submitted as supplementary data (Description, GO terms and fold change). The reader can not see these results. In addition, transcriptional data have to be submitted in public database (e.g. GEO).

-       It is better to report the composition of nutritional solution in the Material and Methods section and not in the supplementary material. The composition could be expressed as molar concentration.

Author Response

Response to Reviewer 3 Comments

Point 1: Introduction: the description of use of RNAseq in order to characterize the plants responses to stresses is unnecessary for the aim of the work. The authors have to improve the introduction with information concerning physiological? (morphological) and molecular aspects of N nutrition in plant with particular attention to low N condition. Some transcriptional works focused on transcriptional responses to low N condition was performed in plant species as arabidopsis and maize.

Response 1: Thank you very much for your valuable comments. I have made a lot of modifications to the introduction of the article, hoping get your approval. Please refer to the introduction of the manuscript. 

Point 2: The authors repeated in the discussion many results already described. In addition, it is more important to discuss the results on the basis of the molecular events involved into N nutrition rather than the number of differentially expressed transcripts as compared to other stresses. Discussion has to be rewritten and improved.

Response 2: Discussion has been major revised. Please refer to the discussion of the manuscript. 

Thank you very much for your valuable comments.

Point 3: Differentially expressed transcripts were not submitted as supplementary data (Description, GO terms and fold change). The reader can not see these results. In addition, transcriptional data have to be submitted in public database (e.g. GEO).

Response 3: Differentially expressed transcripts have been submitted as supplementary data, as shown below:

Table S5: All expression datasets of DEGs under low N stress in leaves. 

Table S6: All expression datasets of DEGs under low N stress in roots. 

The transcriptional data have been deposited in the NCBI Sequence Read Archive (http://www.ncbi.nlm.nih.gov/sra/), and under processing.

Thank you very much for your comments.

Point 4: It is better to report the composition of nutritional solution in the Material and Methods section and not in the supplementary material. The composition could be expressed as molar concentration.

Response 4: These have been revised in the article. Please refer to section 5.2 on page 13 of the manuscript. Thank you very much.

Round  2

Reviewer 2 Report

The authors redo the statistical analysis for DEG screening by using the significant threshold of a false discovery rate (FDR) of 0.05 instead of unadjusted P < 0.05. However, the DEG and related results (in section 2.4 and 2.5) using the FDR adjusted P-value of 0.05 are exact same as results using unadjusted P-value of 0.05 which were written in their previous manuscript. These results are hard to believe since FDR adjusted P-value is more stringent. I would expect less number of DEG genes will be found compared to previous analysis. I suggest the authors include the FDR P-value of 2,265 DEGs found in root and 2083 DEGs found in leaf into supplemental file to convince me.  The DEG results are the most critical part for the whole manuscript because the other analyses (GO, KEGG and qRT-PCR) are all based on the DEG results.

Author Response

Response to Reviewer 2 Comments

Point 1: The authors redo the statistical analysis for DEG screening by using the significant threshold of a false discovery rate (FDR) of 0.05 instead of unadjusted P < 0.05. However, the DEG and related results (in section 2.4 and 2.5) using the FDR adjusted P-value of 0.05 are exact same as results using unadjusted P-value of 0.05 which were written in their previous manuscript. These results are hard to believe since FDR adjusted P-value is more stringent. I would expect less number of DEG genes will be found compared to previous analysis. I suggest the authors include the FDR P-value of 2,265 DEGs found in root and 2083 DEGs found in leaf into supplemental file to convince me. The DEG results are the most critical part for the whole manuscript because the other analyses (GO, KEGG and qRT-PCR) are all based on the DEG results.

Response 1: Thank you very much for your questions and comments. I have uploaded the 2,265 DEGs in root and 2083 DEGs in leaf with P-value as supplementary files, and marked in section 2.3 on page 3 of the manuscript (Table S1 and Table S2).

Please refer to it.

Table S1: All expression datasets of DEGs under low N stress in roots.

Table S2: All expression datasets of DEGs under low N stress in leaves.

Best regards.

Reviewer 3 Report

The new version of the manuscript is improved and suitable for the publication in Plants.

Author Response

Response to Reviewer 3 Comments

Point 1: The new version of the manuscript is improved and suitable for the publication in Plants.

Response 1: Thank you very much for your approval.

Best wishes to you.

Round  3

Reviewer 2 Report

I recommend accepting this manuscript as it is.